# Velocity of *Sargassum* migration in the Caribbean observed with Landsat 8/9 and Sentinel 2 A/B imagery

**Héctor Hernández-Nuñez, Jorge Iván Euán-Avila**[iD]*

Departamento de Recursos del Mar, Centro de Investigación y de Estudios Avanzados, del Instituto Politécnico Nacional, Mérida, Yucatán, México

* jorge.euan@cinvestav.mx

## Abstract

Imagery from Landsat 8/9 (L89) and Sentinel-2 A/B (S2) was employed to monitor the velocity migration of *Sargassum* aggregations. The displacement characteristics of these aggregations offer insights that can inform the formulation of preventive strategies and the planning of harvesting operations for the floating biomass. Images L89 and S2 are sometimes acquired the same day and a few minutes apart. *Sargassum* landmark identification was performed manually on enhanced RGB composite images using quotient indices. A review of images between 2019 and 2023 was performed to select rafts that were distinguishable in both images. Geographic positions were recorded to determine traveled distance, direction, and speed. Pairs of 279 rafts were found on 21 coincident dates. Ninety eight percent of *Sargassum* rafts traveled between 200 m and 1700 m in a time frame of 14 to 26 minutes with an average speed of 0.63 m/s, a standard deviation of 0.24 m/s, a minimum of 0.15 m/s, and a maximum of 1.40m/s. Dominant directions were 34% NW, 23% WNW, 14% NNW, 14% W and 6% N. HYCOM ocean currents showed a positive correlation with *Sargassum* drift, and translation rates are also consistent with surface drifter data. The use of L89 and S2 satellite imagery as an early warning system, in conjunction with current and wind data, may help anticipate the arrival of *Sargassum* in coastal areas.

## Introduction

The massive arrival of pelagic *Sargassum* in the Caribbean has become an environmental concern due to its potential impact on mangroves, seagrasses and coral reefs [1–4]. In addition, the alteration of landscapes and ecosystems through changes in the visual quality of sandy beaches, degradation of water quality and negative changes in the habitats of species for recreational and fishing activities has had a severe impact on the tourism and fishing sectors [2,5]. At the same time, it has been seen as an opportunity to be used for multiple purposes, including animal food, pharmaceutical products, and energy generation [6–8]. The above has motivated studies for its detection and quantification [9–11], as well as to determine the characteristics of its migration in open waters, an aspect of particular interest in decision making for preventive measurements and harvesting operations.

**Data availability statement:** All relevant data are within the manuscript and its Supporting Information files.

**Funding:** The authors have declared that no financial disclosure statement exist.

**Competing interests:** The authors have declared that no competing interests exist

Pelagic *Sargassum* is a marine algae that drifts in the ocean and has highly branched thalli, which can be up to 100 cm in length, with leaves and vesicles that allow the thalli to float either individually and dispersed, or in aggregations of highly variable shape and size [12–14]. Due to their buoyancy characteristics, their distribution is largely at the mercy of currents and winds, forming a variety of aggregations that can extend for tens of kilometers, presenting an opportunity for remote sensing technologies to detect them. In their detection, satellite remote sensing has made a significant contribution, particularly through the use of low spatial resolution but highly synoptic platforms such as MODIS, VIIRS, and Sentinel 3, which have pixel sizes of hundreds of meters and temporal resolutions of hours. Additionally, high-resolution platforms such as L89 and S2 have also been employed, with pixels up to tens of meters and temporal resolutions of days [9,12,15,16].

Due to their orbital design, the Landsat and Sentinel platforms capture scenes very close in time at certain dates [17]. Taking advantage of this capability, the objective of this work is to determine the displacement of *Sargassum* rafts using satellite imagery acquired at short time intervals to determine their displacement, and estimate their speed and direction in a manner analogous to drift buoys.

## Study area

The area selected for study is situated in the Caribbean region, specifically in the Western Caribbean subregion. It is bounded by a rectangle defined by the following corner coordinates: upper-left (19º 43.908' N, 88º 11.070' W), upper-right (19º 28.635' N, 84º 08.2' W), lower-left (16º 37.619' N, 88º 11.070' W), and lower-right (16º 28.081' N, 84º 52.218' W). This area encompasses images from L89 tracks 17 and 18, scenes 47 and 48, as defined by the World Reference System (WRS) of the Landsat program (https://landsat.gsfc.nasa.gov/about/the-worldwide-reference-system/). Additionally, it includes tiles corresponding to S2 (https://hls.gsfc.nasa.gov/products-description/tiling-system/) (Fig 1). The area is strongly influenced by the Yucatan Current and the gyres which occur south of Cuba, as well as by the currents of the Caymanes Basin and those of the Gulf of Honduras [18] (Fig 2). The winds have a regional pattern in which the NE or trade winds dominate, which are derived from the Earth's general atmospheric circulation system due to the Atlantic anticyclone [19]. The continental shelf, in its submerged topography, is relatively short, extending approximately 60 kilometers and reaching a depth of 1,000 meters. In the eastern limit of the study area, at a distance of 350 kilometers from the shoreline, the depth reaches 4,250 meters [20]. The main elevations on the shelf are the island of Cozumel, the Chinchorro Bank and the atolls off the coast of Belize. This coastline from Belize to Cancun in Mexico is an important part of the Mesoamerican reef and *Sargassum* transit to the Gulf of Mexico where, according to estimates made by the Oceanographic Institute of the Gulf and Caribbean Sea in 2022 a 63,000 ton of *Sargassum* have been detected floating in the area [21].

## Method

Floating *Sargassum* rafts have been detected with L89 and S2 images acquired the same day a few minutes apart from one platform to the other; for the most part, the L89 satellites are ahead in time and the S2 satellites follow a few minutes behind. The search for images of the L89 and S2 satellite encounters was conducted between 2019 and 2023. The selection of satellite images initially focused on those acquired by L89. The image pre-processing entailed the application of the Top of Atmosphere (TOA) reflectance. The presence of *Sargassum* rafts was determined visually using an RGB color composite based on band ratios, as proposed by [22]. This composition increases the contrast between the water and the macroalgae due to

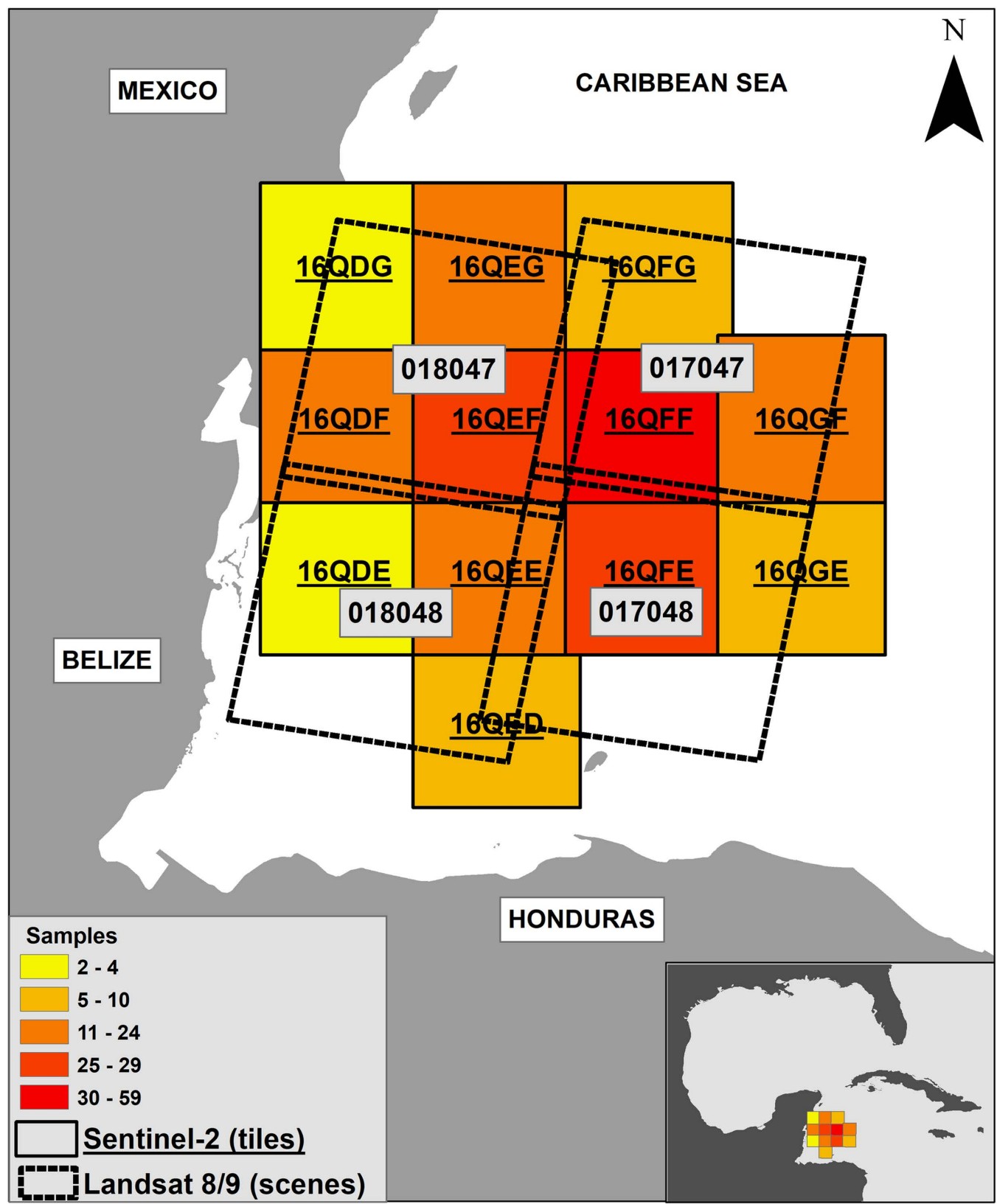

**Fig 1. Study area and samples.** The study area in the Caribbean Sea and the number of *Sargassum* raft samples taken from the L89 and S2 images. Base map from https://noaa.maps.arcgis.com/apps/webappviewer/index.html?id=2be2d19544414752b3088b81ae3f70dd.

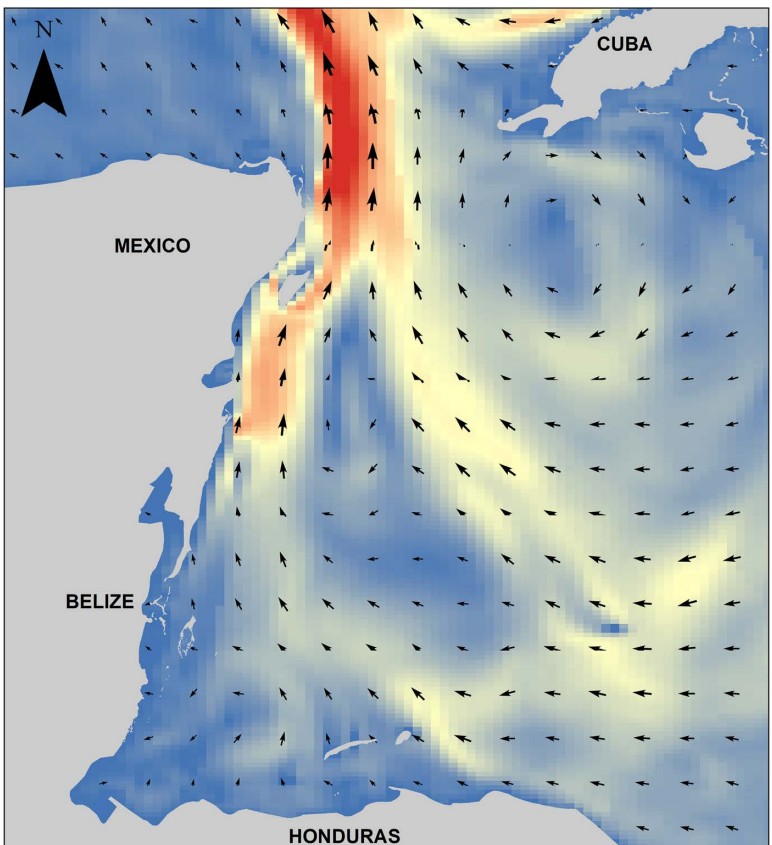

**Fig 2. Ocean currents.** Ocean current pattern model HYCOM (July 20, 2023) https://www.hycom.org/. Base map from https://noaa.maps.arcgis.com/apps/webappviewer/index.html?id=2be2d19544414752b3088b81ae3f70dd, https://www.hycom.org/.

algae photosynthetic activity. The search for S2 scenes from the same date or from one day before or after was then conducted. The same RGB color composite was produced on the S2 images. In a visual analysis, corresponding rafts in both scenes were linked. Pairs of scenes were excluded due to the inability to identify reliable landmarks, which was attributed to the shape alterations observed in the rafts and the masking effect of cloud shadows on *Sargassum* rafts. The linkage of rafts was recorded using the geographic position (UTM- Z16N WGS84) of landmarks on the rafts in both scenes. The preferred landmark location was taken in the central part of the raft, and in others a characteristic feature such as a corner or a discontinuity in the raft perimeter was chosen (Fig 3). Fig 4 illustrates the workflow used to generate the database. The database is provided in the Supporting Information file (S1 File).

The L89 scenes are acquired during a north-south sweep at a local time around 10:25 am (16:25 GMT) (https://www.usgs.gov/faqs/how- can-i-find-acquisition-time-landsat-scene) and around 10:30 am for S2 (https://sentinels.copernicus.eu/web/sentinel/missions/sentinel-2/satellite-description/orbit). The date and time of capture are relative to the center of the scene according to the metadata provided in the S1 File. Time and *Sargassum* traveled distance are used to estimate speed.

The ocean current data (u, v) were downloaded from the Global Ocean Forecasting System (GOFS) 3.1 platform and produced with the HYCOM (HYbrid Coordinate Ocean Model) model (https://www.hycom.org/). The resolution of this model (GOFS 3.1, Global Analysis)

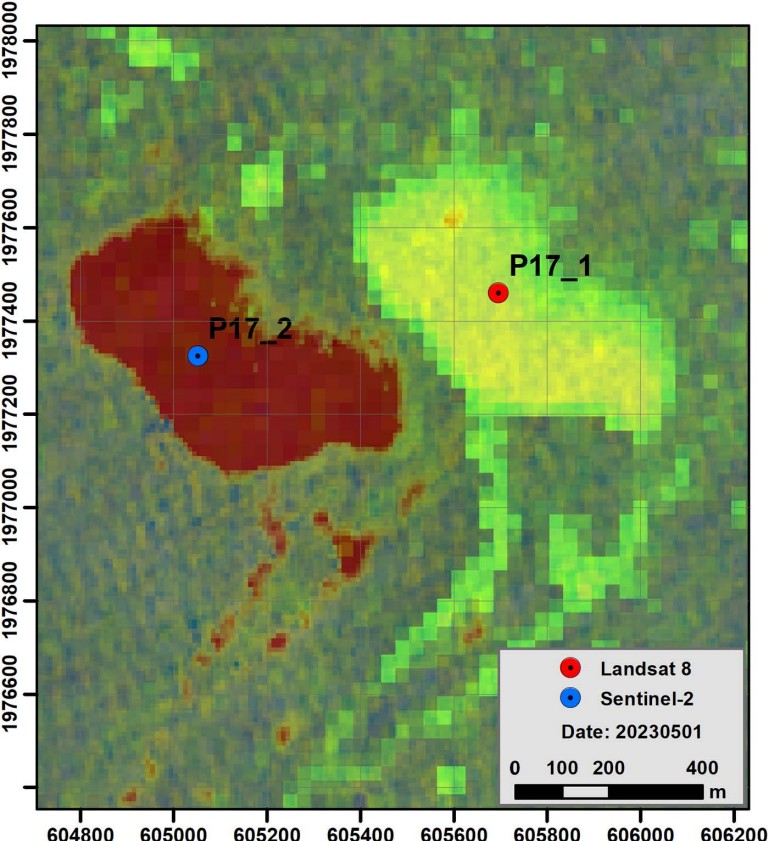

**Fig 3. Rafts selection example.** Point (Lat, Lon) selection on *Sargassum* rafts in Landsat (P17_1 - time 1) and Sentinel 2 (P17_2 - time 2). Images downloaded from https://earthexplorer.usgs.gov/ and https://browser.dataspace.copernicus.eu/?zoom=5&lat=50.16282&lng=20.78613&demSource3D=%22MAPZEN%22&cloudCoverage=30&dateMode=SINGLE.

uses a grid of 0.08° longitude x 0.04° latitude and covers 80S to 90N. The speed and direction values of *Sargassum* in the images were compared with those of the models generated at the time closest to 16:00:00 UTC, the time of the satellite pass. Directions in degrees were converted in 16 standard rose plot with North at 0°. Class frequencies were compared to corroborate how well *Sargassum* and current directions match. Statistical data were processed with InfoStat/L [23].

## Results

### Sargassum

A total of 279 pairs of rafts were identified in L89 and S2 images acquired on the same date during the years 2019, 2021, 2022, and 2023. These findings are presented in Table 1. Total samples of *Sargassum* drift vectors are shown in Fig 5.

Ninety-eight percent of the *Sargassum* rafts traveled between 200 m and 1700 m in a range time of 14 to 26 minutes. Five rafts traveled between 22 and 40 km in 24 hours. Velocities ranged from 0.15 m/s to 1.4 m/s (Table 2). According to the twelve S2 tiles into which the area was divided, Table 3 and Fig 6 show *Sargassum* velocity statistics and Fig 7 illustrates velocity similarities by tile, the QGE (1.39 m/s) and QFG (0.88 m/s) have the highest velocities, while the QEE (0.44 m/s) and QED (0.38 m/s) have the lowest values.

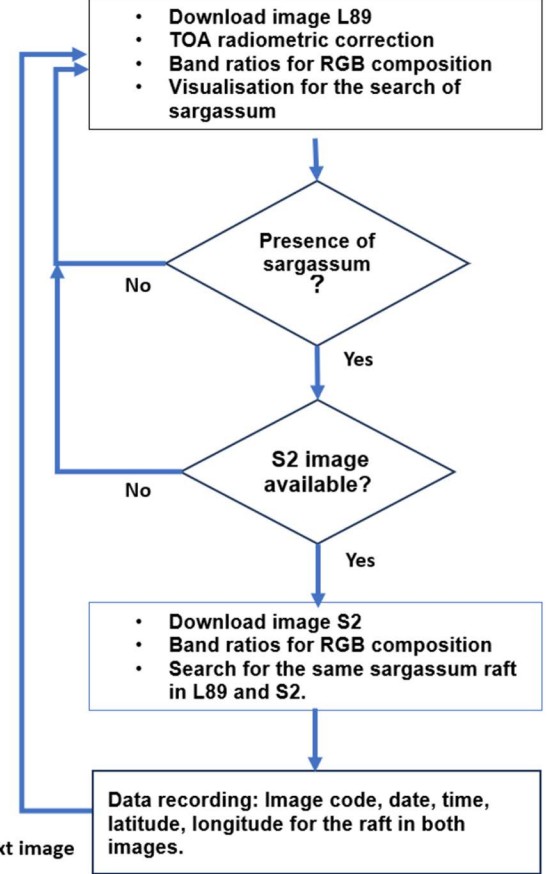

**Fig 4. Workflow.** General steps used for visualization, selection and recording *sargassum* raft data.

**Table 1. Months per year in wich pairs of rafts were found in Landsat 8/9 and Sentinel 2 (A/B) satellites images between the years 2019-2023.**

| Year | Month | Samples |
|---|---|---|
| 2023 | January, February, March, April, May, June, July | 150 |
| 2022 | August, September | 39 |
| 2021 | October | 40 |
| 2020 | n/a | 0 |
| 2019 | January, February, April, May | 50 |
| | Total | 279 |

Direction in which the *Sargassum* moves dominate with 34% towards the NW, 23% WNW, 14% in the NNW direction, 14% W and 6% N (Fig 8). Table 4 shows the statistics of the migration directions within each of the tiles.

## Ocean currents

Statistics of speed of ocean currents were obtained from the closest pixel of HYCOM models the same day satellite images at 16:00 UTC where *Sargassum* was sampled (Tables 5 and 6). The difference of the mean values indicates that the speed of *Sargassum* is 0.13 m/s higher

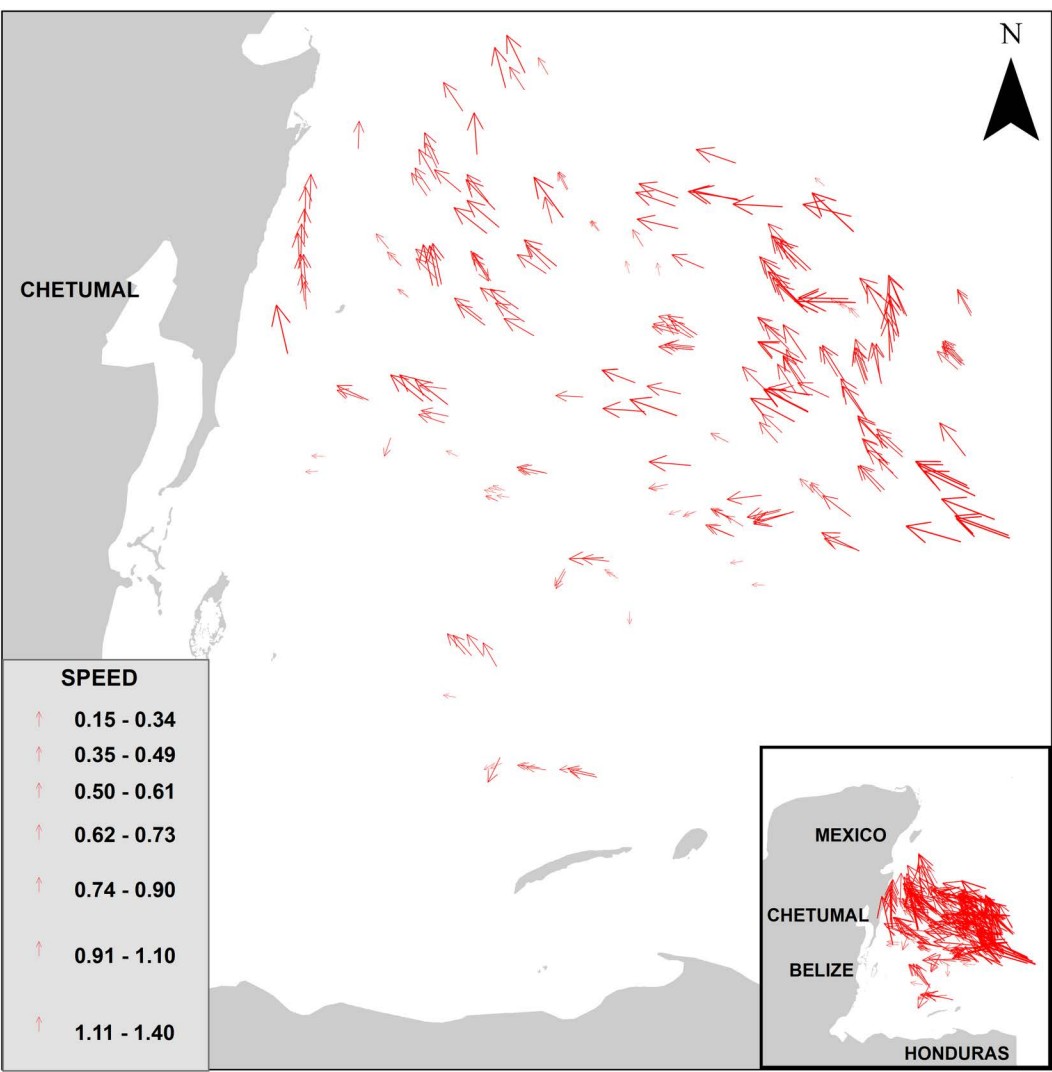

**Fig 5. *Sargassum* drift vectors.** Total samples of *Sargassum* drift speed and direction. Base map from https://noaa.maps. arcgis.com/apps/webappviewer/index.html?id=2be2d19544414752b3088b81ae3f70dd.

Table 2. Total samples *Sargassum* speed statistics (m/s).

| n | Mean | std | Min | Max | Median |
|---|------|-----|-----|-----|--------|
| 279 | 0.64 | 0.24 | .15 | 1.40 | 0.64 |

than the ocean currents and the maximum speed value was also observed in *Sargassum* with 1.4 m/s. The statistics of direction by tile is shown in Table 7. Ocean currents directions show a pattern with dominant directions in NNW (27%), NW (24%) and WNW (13%) (Fig 9).

## Discussion and conclusions

The speeds found are similar to those observed by surface drifters reported by [18,24] with strongest instantaneous speeds above 0.8 m/s located in the Yucatan current with values of up to 1.0 m/s at 20º N latitude. The *Sargassum* speed and the ocean speed modeled by HYCOM is

**Table 3.** *Sargassum* speeds statistics by tiles (m/s).

| Tile | n | Mean | Std | Min | Max | Median |
|------|-----|------|------|------|------|--------|
| QDE | 16 | 0.63 | 0.17 | 0.27 | 0.89 | 0.68 |
| QDF | 21 | 0.58 | 0.18 | 0.28 | 0.94 | 0.56 |
| QDG | 4 | 0.64 | 0.05 | 0.57 | 0.68 | 0.66 |
| QED | 12 | 0.38 | 0.12 | 0.18 | 0.57 | 0.38 |
| QEE | 27 | 0.43 | 0.18 | 0.15 | 0.85 | 0.44 |
| QEF | 35 | 0.64 | 0.19 | 0.27 | 1.03 | 0.65 |
| QEG | 25 | 0.55 | 0.2 | 0.2 | 1.05 | 0.57 |
| QFE | 27 | 0.54 | 0.16 | 0.22 | 0.77 | 0.58 |
| QFF | 71 | 0.71 | 0.19 | 0.26 | 1.12 | 0.69 |
| QFG | 10 | 0.88 | 0.25 | 0.29 | 1.2 | 0.88 |
| QGE | 7 | 1.36 | 0.06 | 1.24 | 1.4 | 1.39 |
| QGF | 24 | 0.71 | 0.11 | 0.55 | 0.85 | 0.71 |

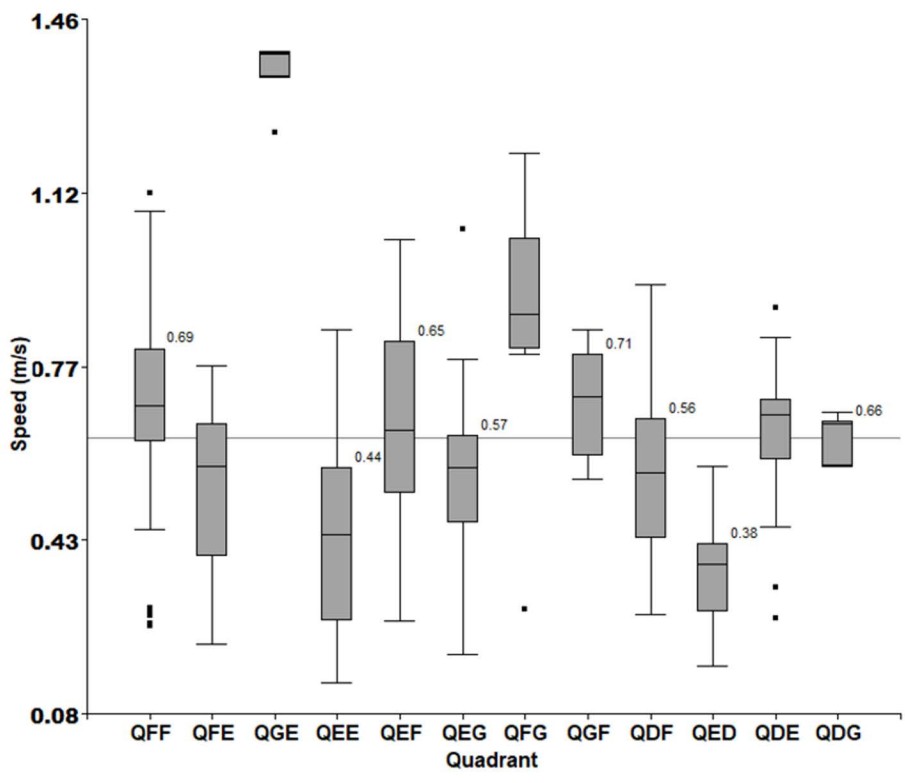

**Fig 6. Speed.** Standard Box-Plot of the *Sargassum* speed by tiles.

positive correlated (R2 = 0.15, p-value = .0001). Similarly the generalized transport of pelagic *Sargassum* towards the NW and its neighbors NNW and WNW are consistent to HYCOM prevalent marine currents in this area. This transport pattern results in a considerable amount of *Sargassum* biomass reaching the coastal areas of the Mexican state of Quintana Roo, leading

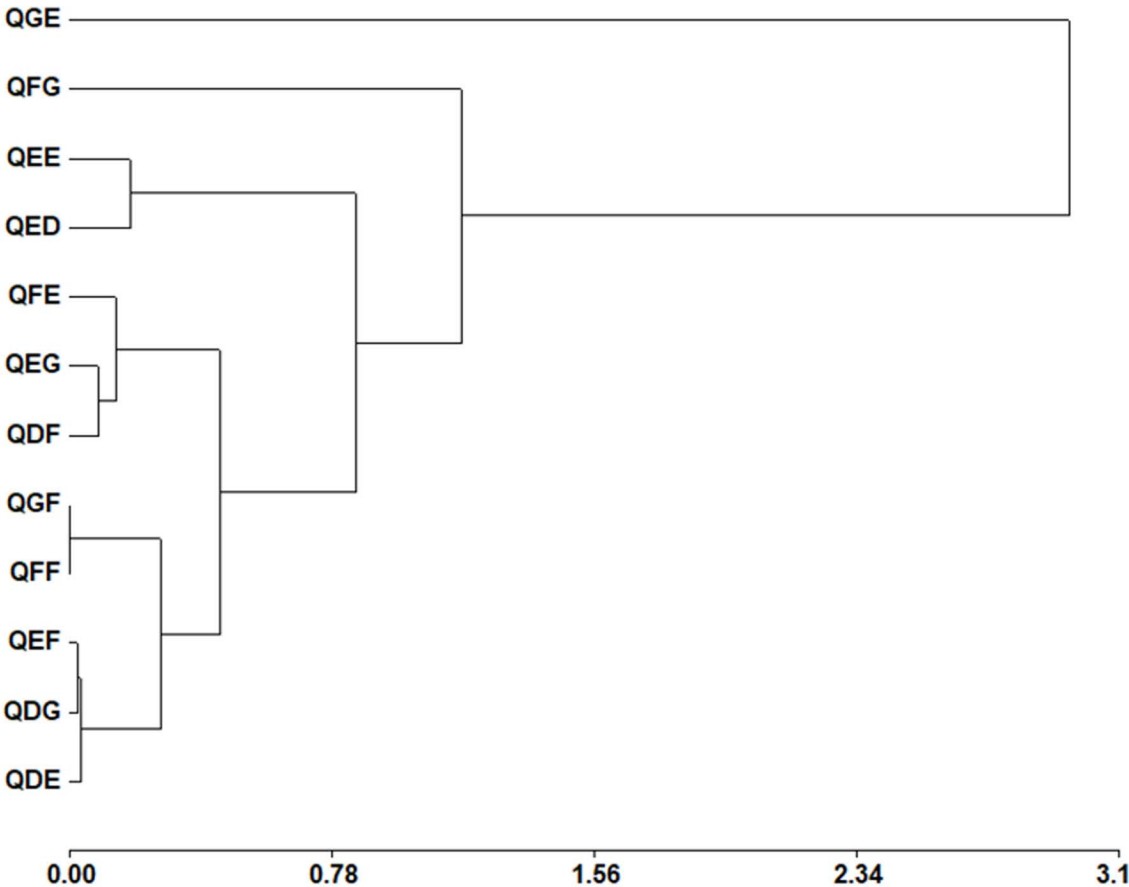

**Fig 7. Speed by tile.** *Sargassum* speed similarities among tiles (Cophenetic correlation = 0.953).

to the formation of extensive accumulations in artificial net barriers situated offshore and on unprotected beaches.

As stated by [25], forecasting the paths of any floating object is inherently challenging due to the multitude of influencing factors that regulate its movement. In this regard, acquiring authentic data, such as those showcased in this study, offers a solid foundation for substantiating the reality of the *Sargassum* movement and can facilitate the evaluation of predictive models.

Monitoring paths 17 and 18 of L89 and corresponding S2 images allowed to detect the presence of *Sargassum* and to estimate a general average speed of 0.63 m/s (2.2 km/h). Together with the dominant observed directions in this work, scenarios of possible arrival times of *Sargassum* rafts at different distances and locations can be constructed. The joint sampling frequency between L89 and S2 with a temporal resolution of three to four days at a distance between 150 and 180 km from the coast will provide an early warning for the presence of *Sargassum*. Moreover, the integration of meteorological data, including wind patterns and ocean currents, from a range of environmental sources, can improve the accuracy of forecasts regarding the potential accumulation of *Sargassum* rafts, thereby reducing the

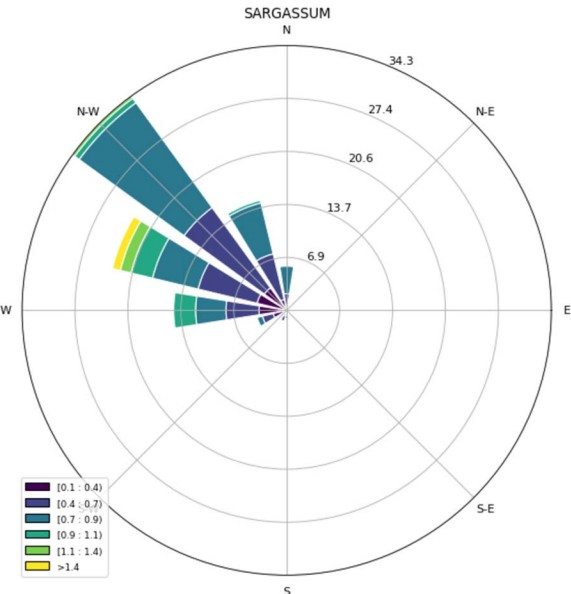

**Fig 8. Wind-rose of velocity for total floating *Sargassum* samples.**

**Table 4. Sargassum directions statistics by tile (0º N).**

| Tile | n | Mean | Std | Min | Max | Median |
|------|------|--------|--------|------|------|--------|
| QDE | 16 | 313.25 | 17.46 | 267 | 325 | 319 |
| QDF | 21 | 297.86 | 77.44 | 1 | 357 | 309 |
| QDG | 4 | 91.25 | 179.17 | 1 | 360 | 2 |
| QED | 12 | 250.17 | 38.52 | 181 | 283 | 271 |
| QEE | 27 | 277.11 | 34.75 | 179 | 330 | 281 |
| QEF | 35 | 307.54 | 35.71 | 150 | 350 | 310 |
| QEG | 25 | 304.56 | 86.61 | 16 | 356 | 328 |
| QFE | 27 | 286.96 | 24.89 | 250 | 318 | 294 |
| QFF | 71 | 304.04 | 20.05 | 264 | 341 | 305 |
| QFG | 10 | 289.2 | 12.84 | 273 | 313 | 286.5 |
| QGE | 7 | 291.86 | 3.63 | 286 | 296 | 291 |
| QGF | 24 | 328.92 | 16.61 | 309 | 358 | 325.5 |

**Table 5. Total samples HYCOM Current speed statistics (m/s).**

| n | Mean | std | Min | Max | Median |
|-----|------|------|------|------|--------|
| 279 | 0.51 | 0.20 | 0.01 | 1.01 | 0.53 |

uncertainty surrounding their arrival and facilitating the formulation of effective coastal management strategies. The infrequent opportunities that platforms L89 and S2 can offer together to obtain close temporal (within minutes) shots, the high presence of clouds in the area, the rapid deformation that rafts undergo over time, and the natural variability of *Sargassum* abundance and distribution throughout the year are the main constraints to integrate a robust data base. In this sense, the spatio-temporal data collected do not have a probabilistic sampling

**Table 6. Current HYCOM speed statistics by tile (m/s).**

| Tile | n | Mean | Std | Min | Max | Median |
|------|-----|------|------|------|------|--------|
| QDE | 16 | 0.48 | 0.05 | 0.4 | 0.56 | 0.5 |
| QDF | 21 | 0.58 | 0.19 | 0.2 | 0.82 | 0.6 |
| QDG | 4 | 0.51 | 0.19 | 0.32 | 0.74 | 0.48 |
| QED | 12 | 0.26 | 0.03 | 0.21 | 0.3 | 0.26 |
| QEE | 27 | 0.27 | 0.13 | 0.06 | 0.54 | 0.27 |
| QEF | 35 | 0.46 | 0.25 | 0.04 | 0.82 | 0.55 |
| QEG | 25 | 0.55 | 0.16 | 0.22 | 0.89 | 0.55 |
| QFE | 27 | 0.59 | 0.08 | 0.42 | 0.73 | 0.59 |
| QFF | 71 | 0.55 | 0.2 | 0.01 | 1.01 | 0.56 |
| QFG | 10 | 0.48 | 0.22 | 0.18 | 0.74 | 0.5 |
| QGE | 7 | 0.83 | 0.09 | 0.68 | 0.93 | 0.85 |
| QGF | 24 | 0.59 | 0.06 | 0.51 | 0.66 | 0.61 |

**Table 7. Currents HYCOM direction statistics by tiles (0º N reference).**

| Tile | n | Mean | Std | Min | Max | Median |
|------|-----|--------|--------|--------|--------|--------|
| QDE | 16 | 316.69 | 23.22 | 259.32 | 341.38 | 320.16 |
| QDF | 21 | 222.13 | 137.46 | 7.82 | 342.07 | 306.68 |
| QDG | 4 | 12.16 | 7.82 | 5.69 | 23.24 | 9.85 |
| QED | 12 | 277.89 | 43.3 | 182.83 | 309.98 | 289.13 |
| QEE | 27 | 290.09 | 35.57 | 181.24 | 341.86 | 291.42 |
| QEF | 35 | 266.53 | 112.2 | 6.43 | 355.3 | 304.38 |
| QEG | 25 | 208.21 | 166.88 | 0.17 | 359.1 | 332.97 |
| QFE | 27 | 309.09 | 35.57 | 247.92 | 355.42 | 325.04 |
| QFF | 71 | 305.53 | 39.49 | 124.04 | 349.18 | 321.71 |
| QFG | 10 | 291.78 | 38.1 | 233.59 | 326.45 | 312.13 |
| QGE | 7 | 274.54 | 4.41 | 267.21 | 279.19 | 274.84 |
| QGF | 24 | 330.85 | 7.3 | 319.38 | 339.4 | 328.76 |

base, but the uncertainty in our measurements can be reduced by adding new observations in future years.

## Supporting information

**S1 File. Landsat 8/9 and Sentinel 2 sampling data and HYCOM ocean current data.** (XLSX)

## Author contributions

**Conceptualization:** Jorge Iván Euan-Avila, Héctor Hernández-Nuñez.

**Data curation:** Héctor Hernández-Nuñez.

**Formal analysis:** Jorge Iván Euan-Avila, Héctor Hernández-Nuñez.

**Investigation:** Jorge Iván Euan-Avila, Héctor Hernández-Nuñez.

**Methodology:** Jorge Iván Euan-Avila, Héctor Hernández-Nuñez.

**Software:** Héctor Hernández-Nuñez.

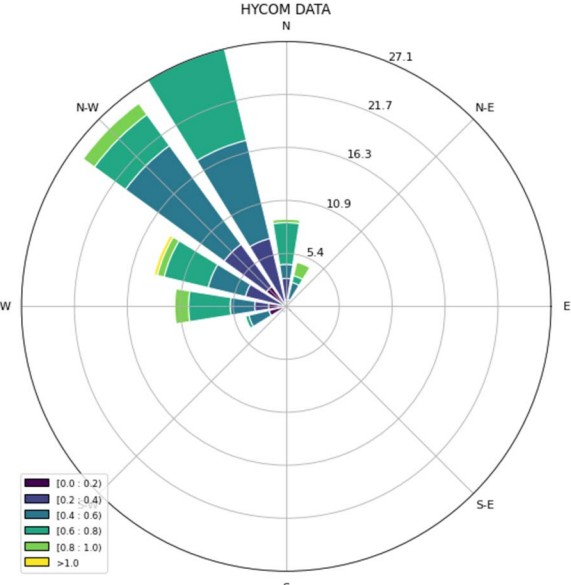

**Fig 9. Wind-rose of velocity for the ocean HYCOM currents.**

**Supervision:** Jorge Iván Euan-Avila.

**Validation:** Jorge Iván Euan-Avila, Héctor Hernández-Nuñez.

**Visualization:** Héctor Hernández-Nuñez.

**Writing – original draft:** Jorge Iván Euan-Avila, Héctor Hernández-Nuñez.

**Writing – review & editing:** Jorge Iván Euan-Avila.

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
