## [Decision Letter · Decision Letter 0]

10 Sep 2024

PONE-D-24-29369Velocity of Sargassum migration in the Caribbean observed with Landsat 8/9 and Sentinel 2 A/B imagery.PLOS ONE

Dear Dr. Euan-Avila,

Thank you for submitting your manuscript to PLOS ONE. After careful consideration, we feel that it has merit but does not fully meet PLOS ONE’s publication criteria as it currently stands. Therefore, we invite you to submit a revised version of the manuscript that addresses the points raised during the review process.

We look forward to receiving your revised manuscript.

Kind regards,

Qing Wang

Academic Editor

PLOS ONE

Journal Requirements: When submitting your revision, we need you to address these additional requirements. 1. Please ensure that your manuscript meets PLOS ONE's style requirements, including those for file naming. The PLOS ONE style templates can be found at https://journals.plos.org/plosone/s/file?id=wjVg/PLOSOne_formatting_sample_main_body.pdf and https://journals.plos.org/plosone/s/file?id=ba62/PLOSOne_formatting_sample_title_authors_affiliations.pdf 2. We note that the grant information you provided in the ‘Funding Information’ and ‘Financial Disclosure’ sections do not match.  When you resubmit, please ensure that you provide the correct grant numbers for the awards you received for your study in the ‘Funding Information’ section. 3. Thank you for stating the following financial disclosure: "This work was supported by Center for Research and Advanced Studies of the National Polytechnic Institute (CINVESTAV) and Consejo Nacional de Humanidades Ciencia y Tecnología (CONAHCYT – SNII)" Please state what role the funders took in the study.  If the funders had no role, please state: ""The funders had no role in study design, data collection and analysis, decision to publish, or preparation of the manuscript."" If this statement is not correct you must amend it as needed. Please include this amended Role of Funder statement in your cover letter; we will change the online submission form on your behalf. 4. We note that your Data Availability Statement is currently as follows: All relevant data are within the manuscript and its Supporting Information files. Please confirm at this time whether or not your submission contains all raw data required to replicate the results of your study. Authors must share the “minimal data set” for their submission. PLOS defines the minimal data set to consist of the data required to replicate all study findings reported in the article, as well as related metadata and methods (https://journals.plos.org/plosone/s/data-availability#loc-minimal-data-set-definition). For example, authors should submit the following data: - The values behind the means, standard deviations and other measures reported;- The values used to build graphs;- The points extracted from images for analysis. Authors do not need to submit their entire data set if only a portion of the data was used in the reported study. If your submission does not contain these data, please either upload them as Supporting Information files or deposit them to a stable, public repository and provide us with the relevant URLs, DOIs, or accession numbers. For a list of recommended repositories, please see https://journals.plos.org/plosone/s/recommended-repositories. If there are ethical or legal restrictions on sharing a de-identified data set, please explain them in detail (e.g., data contain potentially sensitive information, data are owned by a third-party organization, etc.) and who has imposed them (e.g., an ethics committee). Please also provide contact information for a data access committee, ethics committee, or other institutional body to which data requests may be sent. If data are owned by a third party, please indicate how others may request data access.

Reviewers' comments:

Reviewer's Responses to Questions

**Comments to the Author**

1. Is the manuscript technically sound, and do the data support the conclusions?

Reviewer #1: Partly

2. Has the statistical analysis been performed appropriately and rigorously? 

Reviewer #1: No

3. Have the authors made all data underlying the findings in their manuscript fully available?

Reviewer #1: Yes

4. Is the manuscript presented in an intelligible fashion and written in standard English?

Reviewer #1: Yes

5. Review Comments to the Author

Reviewer #1: This is quite an interesting looking at the movement of sargassum rafts across the Caribbean. However the paper lacks some level of rigour. The authors should consider the following comments to make the paper stronger:

1. Clearly this analysis goes to confirm what has alreadybeen established in literature on the role of currents in sargassum transport therefore I reccommend the authors to go beyond just reporting on this link.

2. In their conclusion, the authors stated that 'Together with the dominant observed directions

in this work, scenarios of possible arrival times of Sargassum rafts at different distances and

locations can be constructed". For me with this analysis already done testing this statement could have made the work more rigorous (considering the ratfs that were detected and reported beaching-waht is the linkage?).

3. The methodlogy was not presented with the rigour expected. The authors didn't go into details of how the rafts were segmented and classified beyond just mentioning a semi-automatic segmentation.

4. How were rafts included and excluded in the anlaysis? How were clouds handled?,etc

5. There should be a clear workflow and properly explained.

6. With some further analysis included, clear reccommendations can come out of this work.

7. The english language can also be improved especially with the use of tenses and cosntruction of sentences (some have been pointed out in the pdf.

6. PLOS authors have the option to publish the peer review history of their article (what does this mean? ). If published, this will include your full peer review and any attached files.

**Do you want your identity to be public for this peer review?** For information about this choice, including consent withdrawal, please see our Privacy Policy .

Reviewer #1: No

---

## [Author Response · Author response to Decision Letter 0]

7 Nov 2024

This response is also included in the uploaded file (Response to Reviewers)

Reviewers' comments:

Reviewer's Responses to Questions

Comments to the Author

1. Is the manuscript technically sound, and do the data support the conclusions?

Reviewer #1: Partly

The conclusions presented herein are based on a comprehensive analysis of LANDSAT and Sentinel 2 satellite images (L89 and S2), with the objective of identifying the velocity and direction of concentrations of Sargassum that are in motion. The method, which employs an enhanced image approach, allows for the visual selection of pixels whose geographic positions have been recorded. This enhancing approach provides high contrast among the elements of water, clouds, and Sargasso, facilitating the identification of relevant pixels for the computation of their movement.

2. Has the statistical analysis been performed appropriately and rigorously?

Reviewer #1: No

A probabilistic sampling design is not a viable option given the infrequent opportunities that platforms L89 and S2 can offer together to obtain close temporal (minutes) images. The high prevalence of clouds in the region, the rapid deformation of rafts over time, and the natural variability of Sargassum abundance and distribution over time represent the primary constraints. The data set for standard statistical analysis comprises 279 samples included in the S1 file and covering the period from 2019 to 2023. ________________________________________

3. Have the authors made all data underlying the findings in their manuscript fully available?

Reviewer #1: Yes

Supporting information uploaded in the S1 File provides access to all data underlying the presented statistics. The Excel file contains information on the satellite platform, image name, date, time, raft latitude, raft longitude, raft velocity, raft direction, and HYCOM water velocity and direction.

4. Is the manuscript presented in an intelligible fashion and written in standard English?

Reviewer #1: Yes

The manuscript is presented in an intelligible fashion and written in standard English with attention to language in clear academic terms to be correct and unambiguous.

5. Review Comments to the Author

Reviewer #1: This is quite an interesting looking at the movement of sargassum rafts across the Caribbean. However the paper lacks some level of rigour. The authors should consider the following comments to make the paper stronger:

1. Clearly this analysis goes to confirm what has already been established in literature on the role of currents in sargassum transport therefore I recommend the authors to go beyond just reporting on this link.

As stated by Miron P, Olascoaga MJ, Beron‐Vera FJ, Putman NF, Triñanes J, Lumpkin R, et al. Clustering of Marine‐Debris‐ and Sargassum ‐Like Drifters Explained by Inertial Particle Dynamics. Geophys Res Lett [Internet]. 2020 Oct 16;47(19), forecasting the paths of any floating object is inherently challenging due to the multitude of influencing factors that regulate its movement. In this regard, acquiring hard data, such as those showcased in this study, offers a solid foundation for substantiating the reality of the sargassum movement and can facilitate the evaluation and construction of predictive models to protect beaches and harvest sargassum biomass.

2. In their conclusion, the authors stated that 'Together with the dominant observed directions

in this work, scenarios of possible arrival times of Sargassum rafts at different distances and

locations can be constructed". For me with this analysis already done testing this statement could have made the work more rigorous (considering the ratfs that were detected and reported beaching-waht is the linkage?).

This proposition merits further investigation due to its intrinsic interest. Our current understanding of the project does not encompass the detection and segmentation of the total visible sargassum rafts, the prediction of future occurrences, or the collation of field data in selected marine sites and inshore locations. This is a formidable undertaking that extends beyond the scope of our current objectives.

3. The methodlogy was not presented with the rigour expected. The authors didn't go into details of how the rafts were segmented and classified beyond just mentioning a semi-automatic segmentation.

The method section was updated to include image downloading and radiometric preprocessing. RGB compositions based on reference (22), "Guillaumont B, Bajjouk T, Talec P. Seaweed and remote sensing: a critical review of sensors and data processing." In Round, FE, Chapman, DJ, editors. Progress in phycological research. Biopress Ltd; 1997. pp. 213–82. A workflow Fig 4 illustrate the main steps conducted for the rafts database (S1 File). Furthermore, the specifics of raft selection are outlined. It should be noted that a semi-automatic segmentation process was not employed in the visual analysis to select rafts and was subsequently removed from the workflow.

4. How were rafts included and excluded in the anlaysis? How were clouds handled?,etc

The presence of Sargassum rafts was determined visually using an RGB color composite based on band ratios, as proposed by (22). This composition increases the contrast between the water and the macroalgae due to algae photosynthetic activity, and clouds and shadows are well contrasted from the rafts, thus facilitating their selection.

5. There should be a clear workflow and properly explained.

A workflow figure (Fig 4) was included to provide additional support for the description of the method.

6. With some further analysis included, clear reccommendations can come out of this work.

As stated by (25), forecasting the paths of any floating object is inherently challenging due to the multitude of influencing factors that regulate its movement. In this regard, acquiring authentic data, such as those showcased in this study, offers a solid foundation for substantiating the reality of the sargassum movement and can facilitate the development and evaluation of predictive models.

7. The english language can also be improved especially with the use of tenses and cosntruction of sentences (some have been pointed out in the pdf.

Old Check sentence. Landsat 8/9 (L89) and Sentinel-2 A/B (S2) imagery were used to track velocity migration of Sargassum aggregations. Displacement characteristics provides valuable information for making preventive decisions and planning harvest the biomass.

New sentence. Imagery from Landsat 8/9 (L89) and Sentinel-2 A/B (S2) was employed to monitor the velocity migration of Sargassum aggregations. The displacement characteristics of these aggregations offer insights that can inform the formulation of preventive strategies and the planning of harvesting operations for the biomass

Old Consider re-writing this sentence.A calamity for the tourism and fishing sectors by modifying landscapes and ecosystems through the alteration of the visual quality of sandy beaches, reduction of water quality, and negative changes in the habitats of species for recreational and fishing activities.

New sentence. In addition, the alteration of landscapes and ecosystems through changes in the visual quality of sandy beaches, degradation of water quality and negative changes in the habitats of species for recreational and fishing activities has had a severe impact on the tourism and fishing sectors.

Old Consider re-writing. In their detection, satellite remote sensing through different platforms has contributed monitoring using low spatial resolution but highly synoptic platforms such as MODIS, VIIRS, and Sentinel 3 with pixel sizes of hundreds of meters and temporal resolutions of hours, and high resolution platforms such as L89 and S2 with pixels up to tens of meters and temporal resolutions of days.

New sentence. In their detection, satellite remote sensing has made a significant contribution, particularly through the use of low spatial resolution but highly synoptic platforms such as MODIS, VIIRS, and Sentinel 3, which have pixel sizes of hundreds of meters and temporal resolutions of hours. Additionally, high-resolution platforms such as L89 and S2 have been employed, with pixels up to tens of meters and temporal resolutions of days.

Old Consider re-writing What is located. It is located in the Caribbean, specifically in the Western Caribbean subregion.

New sentence. The area selected for study is situated in the Caribbean region, specifically in the Western Caribbean subregion.

Old Introduce Bounding coordinates. It is bounded by L89 tracks 17 and 18, scenes 47 and 48 according to the world reference system (WRS) of the Landsat program (https://landsat.gsfc.nasa.gov/about/the-worldwide-reference-system/ ) and those tiles corresponding to S2 (https://hls.gsfc.nasa.gov/products-description/tiling-system/ ) (Fig. 1).

New sentence. It is bounded by a rectangle defined by the following corner coordinates: upper-left (19o 43.908' N, 88o 11.070' W), upper-right (19o 28.635' N, 84o 08.2, lower-left (16o 37.619' N, 88o 11.070' W), and lower-right (16o 28.081' N, 84o 52.218' W). This area encompasses images from L89 tracks 17 and 18, scenes 47 and 48, as defined by the World Reference System (WRS) of the Landsat program (https://landsat.gsfc.nasa.gov/about/the-worldwide-reference-system/) . Additionally, it includes tiles corresponding to S2 (https://hls.gsfc.nasa.gov/products-description/tiling-system/).

Old It is not clear which platform. The platform in its submerged topography at 19°N latitude is relatively short, extending about 60km and reaching a depth of 1000 m, at the limits of the study area at a distance of 300km at a depth of 4250 m.

New sentence. The continental shelf, in its submerged topography, is relatively short, extending approximately 60 kilometers and reaching a depth of 1,000 meters. In the eastern limit of the study area, at a distance of 300 kilometers from the shoreline, the depth reaches 4,250 meters.

Old This is not clear. Images where satellites encounters occur were identified between the years 2019 to 2023.

New sentence. The search for images of the L89 and S2 satellite encounters was conducted between 2019 and 2023.

Old This need expatiation. In order to identify the rafts, RGB composite images based on quotient indices were created for both platforms, and a semi-automatic segmentation process was used by the analyst to select and match the rafts.

New text. The selection of satellite images initially focused on those acquired by L89. The image pre-processing entailed the application of the Top of Atmosphere (TOA) reflectance. The presence of Sargassum rafts was determined visually using an RGB color composite based on band ratios, as proposed by Guillaumont (1997). This composition increases the contrast between the water and the macroalgae due to algae photosynthetic activity. The search for S2 scenes from the same date or from one day before or after was then conducted. The same RGB color composite was produced on the S2 images. In a visual analysis, corresponding patches in both scenes were linked. Pairs of scenes were excluded due to the inability to identify reliable landmarks, which was attributed to the shape alterations observed in the rafts and the masking effect of cloud shadows on Sargassum patches. The linkage of rafts was recorded using the geographic position (UTM-WGS84) of landmarks on the rafts in both scenes. The preferred landmark location was taken in the central part of the patch, and in others a characteristic feature such as a corner or a discontinuity in the raft perimeter was chosen. Figure 4 illustrates the workflow used to generate the database, which is provided in the Supporting Information (S1 File).

Old It will be good to include the date in the text. The date and time of capture are relative to the center of the scene according to the metadata.

New text. The date and time (S1 File) of image capture are relative to the center of the scene according to the metadata.

Old Why not determine the center and stick to that for consistency. Otherwise how reliable are you estimates.

Response: The preferred landmark location was taken in the central part of the patch, and in others a characteristic feature such as a corner or a discontinuity in the raft perimeter. This inconsistency was attributed to the inability to identify reliable landmarks, due to the shape alterations observed in the rafts and the masking effect of cloud shadows on Sargassum patches.

Old- check sentence. This transport pattern causes tons of Sargassum biomass to reach on the coastal areas of the Mexican state of Quintana Roo, generating large accumulations in artificial net barriers placed offshore at some distance from the coastline and on the unprotected beaches.

New text. This transport pattern results in a considerable amount of Sargassum biomass reaching the coastal areas of the Mexican state of Quintana Roo, leading to the formation of extensive accumulations in artificial net barriers situated offshore and on unprotected beaches.

6. PLOS authors have the option to publish the peer review history of their article (what does this mean?). If published, this will include your full peer review and any attached files.

Do you want your identity to be public for this peer review? For information about this choice, including consent withdrawal, please see our Privacy Policy.

Reviewer #1: No

---

## [Editor Report · Decision Letter 1]

3 Feb 2025

Velocity of Sargassum migration in the Caribbean observed with Landsat 8/9 and Sentinel 2 A/B imagery.

PONE-D-24-29369R1

Dear Dr. Euan-Avila,

We’re pleased to inform you that your manuscript has been judged scientifically suitable for publication and will be formally accepted for publication once it meets all outstanding technical requirements.

Kind regards,

Qing Wang

Academic Editor

PLOS ONE
---

## [Editor Report · Acceptance letter]

PONE-D-24-29369R1

PLOS ONE

Dear Dr. Euan-Avila,

I'm pleased to inform you that your manuscript has been deemed suitable for publication in PLOS ONE. Congratulations! Your manuscript is now being handed over to our production team.

Kind regards,

on behalf of

Dr. Qing Wang

Academic Editor

PLOS ONE